# Few-Shot Object Detection
# via Association and DIscrimination

**Yuhang Cao**[1]    **Jiaqi Wang**[1,2✉]    **Ying Jin**[1]    **Tong Wu**[1]
**Kai Chen**[2,3]    **Ziwei Liu**[4]    **Dahua Lin**[1,2]

[1]CUHK-SenseTime Joint Lab, The Chinese University of Hong Kong
[2]Shanghai AI Laboratory    [3]SenseTime Research
[4]S-Lab, Nanyang Technological University
{cy020,wj017,jy021,wt020,dhlin}@ie.cuhk.edu.hk
chenkai@sensetime.com    ziwei.liu@ntu.edu.sg

## Abstract

Object detection has achieved substantial progress in the last decade. However, detecting novel classes with only few samples remains challenging, since deep learning under low data regime usually leads to a degraded feature space. Existing works employ a holistic fine-tuning paradigm to tackle this problem, where the model is first pre-trained on all base classes with abundant samples, and then it is used to carve the novel class feature space. Nonetheless, this paradigm is still imperfect. Durning fine-tuning, a novel class may implicitly leverage the knowledge of multiple base classes to construct its feature space, which induces a scattered feature space, hence violating the inter-class separability. To overcome these obstacles, we propose a two-step fine-tuning framework, **F**ew-shot object detection via **A**ssociation and **DI**scrimination (**FADI**), which builds up a discriminative feature space for each novel class with two integral steps. **1)** In the **association** step, in contrast to implicitly leveraging multiple base classes, we construct a compact novel class feature space via explicitly imitating a specific base class feature space. Specifically, we associate each novel class with a base class according to their semantic similarity. After that, the feature space of a novel class can readily imitate the well-trained feature space of the associated base class. **2)** In the **discrimination** step, to ensure the separability between the novel classes and associated base classes, we disentangle the classification branches for base and novel classes. To further enlarge the inter-class separability between all classes, a set-specialized margin loss is imposed. Extensive experiments on standard Pascal VOC and MS-COCO datasets demonstrate that FADI achieves new state-of-the-art performance, significantly improving the baseline in any shot/split by +18.7. Notably, the advantage of FADI is most announced on extremely few-shot scenarios (*e.g.* 1- and 3- shot). Code is available at: https://github.com/yhcao6/FADI

## 1 Introduction

Deep learning has achieved impressive performance on object detection [21, 11, 1] in recent years. However, their strong performance heavily relies on a large amount of labeled training data, which limits the scalability and generalizability of the model in the data scarcity scenarios. In contrast, human visual systems can easily generalize to novel classes with only a few supervisions. Therefore, great interests have been invoked to explore few-shot object detection (FSOD), which aims at training a network from limited annotations of novel classes with the aid of sufficient data of base classes.

---

✉Corresponding author.

35th Conference on Neural Information Processing Systems (NeurIPS 2021).

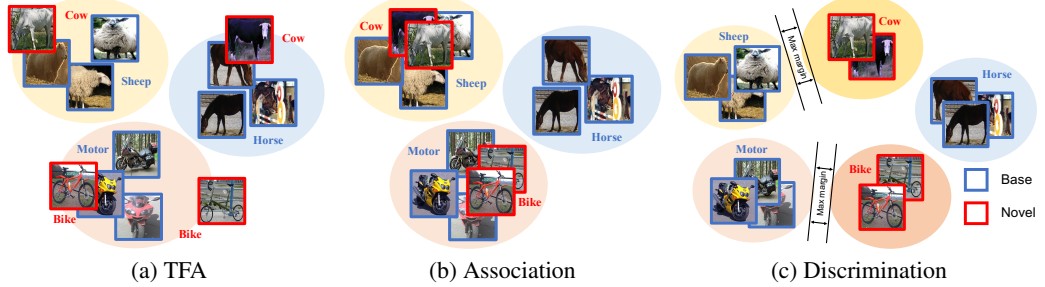

|  (a) TFA | (b) Association | (c) Discrimination |

Figure 1: Conceptually visualization of our FADI. (a) The conventional fine-tuning paradigm, *e.g.*, TFA [28], learns good decision boundaries during the pre-training stage to separate the decision space into several subspaces (rectangles) occupied by different base classes. In the fine-tuning stage, a novel class ('cow') may exploit multiple similar base classes ('sheep' and 'horse') to construct the feature space of itself, which induces a scattered intra-class structure (the feature space of 'cow' across two base classes, 'sheep' and 'horse'). FADI divides the fine-tuning stage into two steps. (b) In the **association** step, to construct a compact intra-class structure, we associate each novel class with a well-learned base class based on their semantic similarity ('cow' is similar to 'sheep', 'motor' is similar to 'bike'). The novel class readily learns to align its intra-class distribution to the associated base class. (c) In the **discrimination** step, to ensure the inter-class separability between novel classes and associated base classes, we disentangle the classification branches for base and novel classes. A set-specialized margin loss is further imposed to enlarge the inter-class separability between all classes.

Various methods have since been proposed to tackle the problem of FSOD, including meta-learning [13, 35, 32], metric learning [14], and fine-tuning [28, 31, 23]. Among them, fine-tuning-based methods are one of the dominating paradigms for few-shot object detection. [28] introduces a simple two-stage fine-tuning approach (TFA). MPSR [31] improves upon TFA [28] via alleviating the problem of scale variation. The recent state-of-the-art method FSCE [23] shows the classifier is more error-prone than the regressor, and introduces the contrastive-aware object proposal encodings to facilitate the classification of detected objects. All these works employ a holistic fine-tuning paradigm, where the model is first trained on all base classes with abundant samples, and then the pre-trained model is fine-tuned on novel classes. Although it exhibits a considerable performance advantage compared with the earlier meta-learning methods, this fine-tuning paradigm is still imperfect. To be specific, the current design of the fine-tuning stage directly extracts the feature representation of a novel class from the network pre-trained on base classes. Therefore, a novel class may exploit the knowledge of multiple similar base classes to construct the feature space of itself. As a result, the feature space of a novel class will have an incompact intra-class structure that scatters across feature spaces of other classes, breaking the inter-class separability, hence leading to classification confusion, as shown in Figure 1a.

To overcome these obstacles, we propose a two-step fine-tuning framework, **F**ew-shot object detection via **A**ssociation and **DI**scrimination (**FADI**), which constructs a discriminable feature space for each novel class with two integral steps, **association** and **discrimination**. Specifically, in the **association** step, as shown in Figure 1b, to construct a compact intra-class structure, we associate each novel class with a well-trained base class based on their underlying semantic similarity. The novel class readily learns to align its feature space to the associated base class, thus naturally becomes separable from the remaining classes. In the **discrimination** step, as shown in Figure 1c, to ensure the separability between the novel classes and associated base classes, we disentangle the classification branches for base and novel classes to reduce the ambiguity in the feature space induced by the association step. To further enlarge the inter-class separability between all classes, a set-specialized margin loss is applied. To this end, the fine-tuning stage is divided into two dedicated steps, and together complement each other.

Extensive experimental results have validated the effectiveness of our approach. We gain significant performance improvements on the Pascal VOC [7] and COCO [18] benchmarks, especially on the extremely few-shot scenario. Specifically, without bells and whistles, FADI improves the TFA [28] baseline by a significant margin in any split and shot with up to +18.7 mAP, and push the envelope of the state-of-the-art performance by 2.5, 4.3, 2.8 and 5.6, 7.8, 1.6 for shot $K = 1, 2, 3$ on novel split-1 and split-3 of Pascal VOC dataset, respectively.

## 2   Related Work

**Few-Shot Classification**   Few-Shot Classification aims to recognize novel instances with abundant base samples and a few novel samples. Metric-based methods address the few-shot learning by learning to compare, different distance formulations [26, 22, 24] are adopted. Initialization-based methods [9, 15] learn a good weight initialization to promote the adaption to unseen samples more effectively. Hallucination-based methods introduce hallucination techniques [10, 29] to alleviate the shortage of novel data. Recently, researchers find out that the simple pre-training and fine-tuning framework [4, 6] can compare favorably against other complex algorithms.

**Few-Shot Object Detection**   As an emerging task, FSOD is less explored than few-shot classification. Early works mainly explore the line of meta-learning [13, 35, 32, 14, 36, 37, 8], where a meta-learner is introduced to acquire class agnostic meta knowledge that can be transferred to novel classes. Later, [28] introduces a simple two-stage fine-tuning approach (TFA), which significantly outperforms the earlier meta-learning methods. Following this framework, MPSR [31] enriches object scales by generating multi-scale positive samples to alleviate the inherent scale bias. Recently, FSCE [23] shows in FSOD, the classifier is more error-prone than the regressor and introduces the contrastive-aware object proposal encodings to facilitate the classification of detected objects. Similarly, FADI also aims to promote the discrimination capacity of the classifier. But unlike previous methods that directly learn the classifier by implicitly exploiting the base knowledge, motivated by the works of [34, 33], FADI explicitly associates each novel class with a semantically similar base class to learn a compact intra-class distribution.

**Margin Loss**   Loss function plays an important role in the field of recognition tasks. To enhance the discrimination power of traditional softmax loss, different kinds of margin loss are proposed. SphereFace [19] introduces a multiplicative margin constrain in a hypersphere manifold. However, the non-monotonicity of cosine function makes it difficult for stable optimization, CosFace [27] then proposed to further normalize the feature embedding and impose an additive margin in the cosine space. ArcFace [5] moves the additive cosine margin into the angular space to obtain a better discrimination power and more stable training. However, we find these margin losses are not directly applicable under data-scarce settings as they equally treat different kinds of samples but ignore the inherent bias of the classifier towards base classes. Hence we propose a set-specialized margin loss that takes the kind of samples into consideration which yields significantly better performance.

## 3   Our Approach

In this section, we first review the preliminaries of few-shot object detection setting and the conventional two-stage fine-tuning framework. Then we introduce our method that tackles few-shot object detection via association and discrimination (FADI).

### 3.1   Preliminaries

In few-shot detection, the training set is composed of a base set $D^B = \{x_i^B, y_i^B\}$ with abundant data of classes $C^B$, and a novel set $D^N = \{x_i^N, y_i^N\}$ with few-shot data of classes $C^N$, where $x_i$ and $y_i$ indicate training samples and labels, respectively. The number of objects for each class in $C^N$ is $K$ for $K$-shot detection. The model is expected to detect objects in the test set with classes in $C^B \cup C^N$.

Fine-tuning-based methods are the current one of the leading paradigms for few-shot object detection, which successfully adopt a simple two-stage training pipeline to leverage the knowledge of base classes. TFA [28] is a widely adopted baseline of fine-tuning-based few-shot detectors. In the base training stage, the model is trained on base classes with sufficient data to obtain a robust feature representation. In the novel fine-tuning stage, the pre-trained model on base classes is then fine-tuned on a balanced few-shot set which comprises both base and novel classes ($C_B \cup C_N$). Aiming at preventing over-fitting during fine-tuning, only the box predictor, *i.e.*, classifier and regressor, are updated to fit the few-shot set. While the feature extractor, *i.e.*, other structures of the network, are frozen [28] to preserve the pre-trained knowledge on the abundant base classes.

Although the current design of fine-tuning stage brings considerable gains on few-shot detection, we observe that it may induce a scattered feature space on novel class, which violates the inter-class

Figure 2: **Method overview**. There are two steps in FADI: association and discrimination. To construct a compact intra-class structure, the association step aligns the feature distribution of each novel class with a well-learned base class based on their semantic similarity. To ensure inter-class separability, the discrimination step disentangles classification branches for base and novel classes and imposes a set-specialized margin loss.

separability and leads to confusion of classification. Towards this drawback, we proposes few-shot object detection via **association** and **discrimination** (FADI), which divides the fine-tuning stage of TFA into a two-step **association** and **discrimination** pipelines. In the **association** step (Sec. 3.2), to construct a compact intra-class distribution, we associate each novel class with a base class based on their underlying semantic similarity. The feature representation of the associated base class is explicitly learned by the novel class. In the **discrimination** step (Sec. 3.3), to ensure the inter-class separability, we disentangle the base and novel branches and impose a set-specialized margin loss to train a more discriminative classifier for each class.

## 3.2   Association Step

In the base training stage, the base model is trained on the abundant base data $D^B$ and its classifier learns a good decision boundary (see Figure 1) to separate the whole decision space into several subspaces that are occupied by different base classes. Therefore, if a novel class can align the feature distribution of a base class, it will fall into the intra-class distribution of the associated base class, and be naturally separable from the other base classes. And if two novel classes are assigned to different base classes, they will also become separable from each other.

To achieve this goal, we introduce a new concept named association, which pairs each novel class to a similar base class by **semantic similarity**. After then, the feature distribution of the novel class is aligned with the associated base class via **feature distribution alignment**.

**Similarity Measurement**   In order to ease the difficulty of feature distribution alignment, given a novel class $C_i^N$ and a set of base classes $C^B$, we want to associate $C_i^N$ to the most similar base class in $C^B$. An intuitive way is to rely on visual similarity between feature embeddings. However, the embedding is not representative for novel classes under data-scarce scenarios. Thus, we adopt WordNet [20] as an auxiliary to describe the semantic similarity between classes. WordNet is an English vocabulary graph, where nodes represent lemmas or synsets and they are linked according to their relations. It incorporates rich lexical knowledge which benefits the association. Lin Similarity [16] is used to calculate the class-to-class similarity upon WordNet which is given by:

$$\text{sim}(C_i^N, C_j^B) = \frac{2 \cdot \text{IC}(\text{LCS}(C_i^N, C_j^B))}{\text{IC}(C_i^N) + \text{IC}(C_j^B)}, \tag{1}$$

where LCS denotes the lowest common subsumer of two classes in lexical structure of WordNet. IC, *i.e.*, information content, is the probability to encounter a word in a specific corpus. SemCor Corpus is adopted to count the word frequency here. We take the maximum among all base classes to obtain

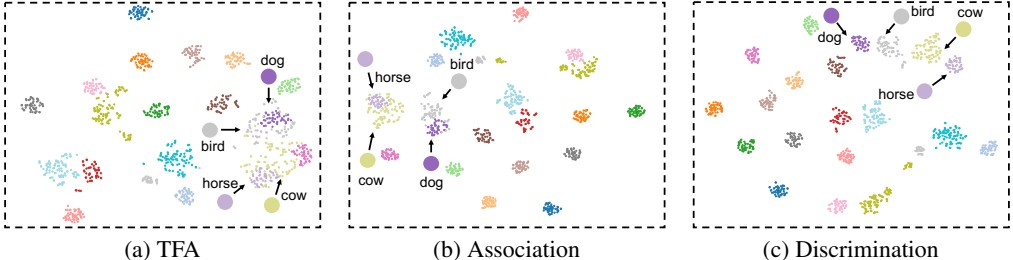

|       |            |                  |
|-------|------------|------------------|
| (a) TFA | (b) Association | (c) Discrimination |

Figure 3: t-SNE here shows the distribution of feature after $FC_2/FC_2'$ from 200 randomly selected images on PASCAL VOC, 'horse' and 'dog' are base classes, 'cow' and 'bird' are novel classes, respectively. The feature space learned by FADI has a more compact intra-class structure and larger inter-class separability.

the associated base class $C_{j \to i}^B$ where $j \to i$ means the base class $C_j^B$ is assigned to the novel class $C_i^N$.

$$C_{j \to i}^B \leftarrow \operatorname*{argmax}_{j \in |C^B|} \operatorname{sim}(C_i^N, C_j^B). \tag{2}$$

To this end, novel class set $C^N$ is associated with a subset of base class $C^{B \to N} \subset C^B$.

**Feature Distribution Alignment** After obtaining the associated base class for each novel class, given a sample $x_i^N$ of novel class $C_i^N$, it is associated with a pseudo label $y_j^B$ of the assigned base class $C_{j \to i}^B$. We design a pseudo label training mechanism to directly align the feature distribution of the novel class with the assigned base class, as follows.

$$\min_{\mathcal{W}_{asso}^N} \mathcal{L}_{cls}(y_j^B, f(\mathbf{z}_i^N; \widetilde{\mathcal{W}}_{cls}^B)), \text{ where } \mathbf{z}_i^N = g(\phi(x_i^N; \widetilde{\mathcal{W}}_{pre}^B); \mathcal{W}_{asso}^N), \tag{3}$$

where $\widetilde{\mathcal{W}}$ means the weights are frozen. Thus, $f(\cdot; \widetilde{\mathcal{W}}_{cls}^B)$ and $\phi(\cdot; \widetilde{\mathcal{W}}_{pre}^B)$ indicate the classifier (one fc layer) and the feature extractor (main network structures) with frozen weights and are pre-trained on base classes, and $g(\cdot; \mathcal{W}_{asso}^N)$ means an intermediate structure (one or more fc layers) to align the feature distribution via updating the weights $\mathcal{W}_{asso}^N$. By assigning pseudo labels and freezing the classifier, this intermediate structure learns to align the feature distribution of the novel class to the associated base class. The main network structures $\phi(\cdot; \widetilde{\mathcal{W}}_{pre}^B)$ is also fixed to keep the pre-trained knowledge from base classes.

As shown in Figure 2, we use the same RoI head structure of Faster R-CNN [21], but we remove the regressor to reduce it to a pure classification problem. During training, we freeze all parameters except the second linear layer $FC_2'$, which means $g(\cdot; \mathcal{W}_{asso}^N)$ is a single fc layer. We then construct a balanced training set with $K$ shots per class. It is noted we discard the base classes that are associated with novel classes in this step. And the labels of novel classes are replaced by their assigned pseudo labels. As a result, the supervision will enforce the classifier to identify samples of the novel class $C_i^N$ as the assigned base class $C_{j \to i}^B$, which means the feature representation of novel classes before the classifier gradually shifts toward their assigned base classes. As shown in Figure 3b, the t-SNE [25] visualization confirms the effectiveness of our distribution alignment. After the association step, the feature distribution of two associated pairs ("bird" and "dog"; "cow" and "horse") are well aligned.

### 3.3 Discrimination Step

As shown in Figure 3b, after the association step, the feature distribution of each novel class is aligned with the associated base class. Therefore, this novel class will have a compact intra-class distribution and be naturally distinguishable from other classes. However, the association step inevitably leads to confusion between the novel class and its assigned base class. To tackle this problem, we introduce a discrimination step that disentangles the classification branches for base and novel classes. A set-specialized margin loss is further applied to enlarge the inter-class separability.

**Disentangling** Given a training sample $x_i$ with label $y_i$, we disentangle the classification branches for base and novel classes as follows,

$$\min_{\mathcal{W}_{cls}^B, \mathcal{W}_{cls}^N} \mathcal{L}_{cls}(y_i, [\mathbf{p}^B, \mathbf{p}^N]), \text{ where}$$

$$\mathbf{p}^B = f(g(\mathbf{q}; \widetilde{\mathcal{W}}_{origin}^B); \mathcal{W}_{cls}^B), \mathbf{p}^N = f(g(\mathbf{q}; \widetilde{\mathcal{W}}_{asso}^N); \mathcal{W}_{cls}^N), \mathbf{q} = \phi(x_i; \widetilde{\mathcal{W}}_{pre}^B), \quad (4)$$

where $f(\cdot; \mathcal{W}_{cls}^B)$, $f(\cdot; \mathcal{W}_{cls}^N)$ are the classifiers for base and novel classes, respectively. $g(\cdot, \widetilde{\mathcal{W}}_{origin}^B)$, $g(\cdot, \widetilde{\mathcal{W}}_{asso}^N)$ are the last fc layer with frozen weights for base and novel classes, respectively. As shown in Figure 2, we disentangle the classifiers and the last fc layers ($FC_2$ and $FC_2{}'$) for base and novel classes. $FC_2$ and $FC_2{}'$ load the original weights $\widetilde{\mathcal{W}}_{origin}^B$ that are pre-trained with base classes and the weights $\widetilde{\mathcal{W}}_{asso}^N$ after association step, respectively. They are frozen in the discrimination step to keep their specific knowledge for base and novel classes. Therefore, $FC_2$ and $FC_2{}'$ are suitable to deal with base classes and novel classes, respectively. We attach the base classifier $f(\cdot; \mathcal{W}_{cls}^B)$ to $FC_2$, and the novel classifier $f(\cdot; \mathcal{W}_{cls}^N)$ to $FC_2{}'$. The base classifier is a $|C_B|$-way classifier. The novel classifier is a $(|C_N|+1)$-way classifier since we empirically let the novel classifier be also responsible for recognizing background class $C_0$. The prediction $\mathbf{p}^B$ and $\mathbf{p}^N$ from these two branches will be concatenated to yield the final $(|C_B| + |C_N| + 1)$-way prediction $[\mathbf{p}^B, \mathbf{p}^N]$.

**Set-Specialized Margin Loss** Besides disentangling, we further propose a set-specialized margin loss to alleviate the confusion between different classes. Different from previous margin losses [19, 27, 5] that directly modify the original CE loss, we introduce a margin loss as an auxiliary loss for the classifier. Given an $i$-th training sample of label $y_i$, we adopt cosine similarity to formulate the logits prediction, which follows the typical conventions in few-shot classification and face recognition [27].

$$p_{y_i} = \frac{\tau \cdot \mathbf{x}^T \mathcal{W}_{y_i}}{||\mathbf{x}|| \cdot ||\mathcal{W}_{y_i}||}, \quad s_{y_i} = \frac{e^{p_{y_i}}}{\sum_{j=1}^{C} e^{p_j}}, \quad (5)$$

where $\mathcal{W}$ is the weight of the classifier, $\mathbf{x}$ is the input feature and $\tau$ is the temperature factor. We try to maximize the margin of decision boundary between $C_{y_i}$ and any other class $C_{j, j \neq y_i}$, as follows,

$$\mathcal{L}_{m_i} = \sum_{j=1, j \neq y_i}^{C} -\log((s_{y_i} - s_j)^+ + \epsilon), \quad (6)$$

where $s_{y_i}$ and $s_j$ are classification scores on class $C_{y_i}$ and $C_{j, j \neq y_i}$, and $\epsilon$ is a small number ($1e^{-7}$) to keep numerical stability.

In the scenario of few-shot learning, there exists an inherent bias that the classifier tends to predict higher scores on base classes, which makes the optimization of margin loss on novel classes becomes more difficult. And the number of background (negative) samples dominates the training samples, thus we may suppress the margin loss on background class $C_0$.

Towards the aforementioned problem, it is necessary to introduce the set-specialized handling of different set of classes into the margin loss. Thanks to adopting margin loss as an auxiliary benefit, our design can easily enable set-specialized handling of different sets of classes by simply re-weighting the margin loss value:

$$\mathcal{L}_m = \sum_{\{i|y_i \in C^B\}} \alpha \cdot \mathcal{L}_{m_i} + \sum_{\{i|y_i \in C^N\}} \beta \cdot \mathcal{L}_{m_i} + \sum_{\{i|y_i = C^0\}} \gamma \cdot \mathcal{L}_{m_i}, \quad (7)$$

where $\alpha, \beta, \gamma$ are hyper-parameters controlling the margin of base samples, novel samples and negative samples, respectively. Intuitively, $\beta$ is larger than $\alpha$ because novel classes are more challenging, and $\gamma$ is a much smaller value to balance the overwhelming negative samples. Finally, the loss function of the discrimination step is shown as in Eq. 8

$$\mathcal{L}_{ft} = \mathcal{L}_{cls} + \mathcal{L}_m + 2 \cdot \mathcal{L}_{reg}, \quad (8)$$

where $\mathcal{L}_{cls}$ is a cross-entropy loss for classification, $\mathcal{L}_{reg}$ is a smooth-L1 loss for regression, and $\mathcal{L}_m$ is the proposed set-specialized margin loss. Since our margin loss increases the gradients on the classification branch, we scale $\mathcal{L}_{reg}$ by a factor of 2 to keep the balance of the two tasks. The overall loss takes the form of multi-task learning to jointly optimize the model.

| Method / Shot | Backbone | Novel Split 1 | | | | | Novel Split 2 | | | | | Novel Split 3 | | | | |
|---|---|---|---|---|---|---|---|---|---|---|---|---|---|---|---|---|
| | | 1 | 2 | 3 | 5 | 10 | 1 | 2 | 3 | 5 | 10 | 1 | 2 | 3 | 5 | 10 |
| LSTD [2] | VGG-16 | 8.2 | 1.0 | 12.4 | 29.1 | 38.5 | 11.4 | 3.8 | 5.0 | 15.7 | 31.0 | 12.6 | 8.5 | 15.0 | 27.3 | 36.3 |
| YOLOv2-ft [30] | | 6.6 | 10.7 | 12.5 | 24.8 | 38.6 | 12.5 | 4.2 | 11.6 | 16.1 | 33.9 | 13.0 | 15.9 | 15.0 | 32.2 | 38.4 |
| †FSRW [13] | YOLO V2 | 14.8 | 15.5 | 26.7 | 33.9 | 47.2 | 15.7 | 15.3 | 22.7 | 30.1 | 40.5 | 21.3 | 25.6 | 28.4 | 42.8 | 45.9 |
| †MetaDet [30] | | 17.1 | 19.1 | 28.9 | 35.0 | 48.8 | 18.2 | 20.6 | 25.9 | 30.6 | 41.5 | 20.1 | 22.3 | 27.9 | 41.9 | 42.9 |
| †RepMet [14] | InceptionV3 | 26.1 | 32.9 | 34.4 | 38.6 | 41.3 | 17.2 | 22.1 | 23.4 | 28.3 | 35.8 | 27.5 | 31.1 | 31.5 | 34.4 | 37.2 |
| FRCN-ft [30] | | 13.8 | 19.6 | 32.8 | 41.5 | 45.6 | 7.9 | 15.3 | 26.2 | 31.6 | 39.1 | 9.8 | 11.3 | 19.1 | 35.0 | 45.1 |
| FRCN+FPN-ft [28] | FRCN-R101 | 8.2 | 20.3 | 29.0 | 40.1 | 45.5 | 13.4 | 20.6 | 28.6 | 32.4 | 38.8 | 19.6 | 20.8 | 28.7 | 42.2 | 42.1 |
| †MetaDet [30] | | 18.9 | 20.6 | 30.2 | 36.8 | 49.6 | 21.8 | 23.1 | 27.8 | 31.7 | 43.0 | 20.6 | 23.9 | 29.4 | 43.9 | 44.1 |
| †Meta R-CNN [35] | | 19.9 | 25.5 | 35.0 | 45.7 | 51.5 | 10.4 | 19.4 | 29.6 | 34.8 | 45.4 | 14.3 | 18.2 | 27.5 | 41.2 | 48.1 |
| TFA w/ fc [28] | | 36.8 | 29.1 | 43.6 | 55.7 | 57.0 | 18.2 | 29.0 | 33.4 | 35.5 | 39.0 | 27.7 | 33.6 | 42.5 | 48.7 | 50.2 |
| TFA w/ cos [28] | | 39.8 | 36.1 | 44.7 | 55.7 | 56.0 | 23.5 | 26.9 | 34.1 | 35.1 | 39.1 | 30.8 | 34.8 | 42.8 | 49.5 | 49.8 |
| MPSR [31] | FRCN-R101 | 41.7 | - | 51.4 | 55.2 | 61.8 | 24.4 | - | 39.2 | 39.9 | 47.8 | 35.6 | - | 42.3 | 48.0 | 49.7 |
| SRR-FSD [38] | | 47.8 | 50.5 | 51.3 | 55.2 | 56.8 | **32.5** | **35.3** | 39.1 | 40.8 | 43.8 | 40.1 | 41.5 | 44.3 | 46.9 | 46.4 |
| FSCE [23] | | 44.2 | 43.8 | 51.4 | **61.9** | **63.4** | 27.3 | 29.5 | **43.5** | **44.2** | **50.2** | 37.2 | 41.9 | 47.5 | 54.6 | 58.5 |
| FADI (Ours) | | **50.3** | **54.8** | **54.2** | 59.3 | 63.2 | 30.6 | 35.0 | 40.3 | 42.8 | 48.0 | **45.7** | **49.7** | **49.1** | **55.0** | **59.6** |

Table 1: Performance (novel AP50) on PASCAL VOC dataset. † denotes meta-learning-based methods.

| shot | nAP | | nAP50 | | nAP75 | |
|---|---|---|---|---|---|---|
| | TFA | FADI | TFA | FADI | TFA | FADI |
| 1 | 3.4 | **5.7** | 5.8 | **10.4** | 3.8 | **6.0** |
| 2 | 4.6 | **7.0** | 8.3 | **13.1** | 4.8 | **7.0** |
| 3 | 6.6 | **8.6** | 12.1 | **15.8** | 6.5 | **8.3** |
| 5 | 8.3 | **10.1** | 15.3 | **18.6** | 8.0 | **9.7** |
| 10 | 10.0 | **12.2** | 19.1 | **22.7** | 9.3 | **11.9** |
| 30 | 13.7 | **16.1** | 24.9 | **29.1** | 13.4 | **15.8** |

| Method | nAP | | nAP75 | |
|---|---|---|---|---|
| | 10 | 30 | 10 | 30 |
| †FSRW [13] | 5.6 | 9.1 | 4.6 | 7.6 |
| †MetaDet [30] | 7.1 | 11.3 | 5.9 | 10.3 |
| †Meta R-CNN [35] | 8.7 | 12.4 | 6.6 | 10.8 |
| MPSR [31] | 9.8 | 14.1 | 9.7 | 14.2 |
| SRR-FSD [38] | 11.3 | 14.7 | 9.8 | 13.5 |
| FSCE [23] | 11.9 | **16.4** | 10.5 | **16.2** |
| Ours (FADI) | **12.2** | 16.1 | **11.9** | 15.8 |

(a) Comparison with baseline TFA  (b) Comparison with latest methods.

Table 2: Performance on MS COCO dataset. † denotes meta-learning-based methods. nAP means novel AP.

# 4 Experiments

## 4.1 Datasets and Evaluation Protocols

We conduct experiments on both PASCAL VOC (07 + 12) [7] and MS COCO [18] datasets. To ensure fair comparison, we strictly follow the data split construction and evaluation protocol used in [13, 28, 23]. PASCAL VOC contains 20 categories, and we consider the same 3 base/novel splits with TFA [28] and refer them as Novel Split 1, 2, 3. Each split contains 15 base categories with abundant data and 5 novel categories with $K$ annotated instances for $K = 1, 2, 3, 5, 10$. We report AP50 of novel categories (**nAP50**) on VOC07 test set. For MS COCO, 20 classes that overlap with PASCAL VOC are selected as novel classes, and the remaining 60 classes are set as base ones. Similarly, we evaluate our method on shot 1, 2, 3, 5, 10, 30 and the standard COCO-style ap metric is adopted.

## 4.2 Implementation Details

We implement our methods based on MMDetection [3]. Faster-RCNN [21] with Feature Pyramid Network [17] and ResNet-101 [12] are adopted as base model. Detailed settings are described in the supplementary material.

## 4.3 Benchmarking Results

**Comparison with Baseline Methods** To show the effectiveness of our method, we first make a detailed comparison with TFA since our method is based on it. As shown in Table 1, FADI outperforms TFA by a large margin in any shot and split on PASCAL VOC benchmark. To be specific, FADI improves TFA by 10.5, 18.7, 9.5, 3.6, 7.2 and 7.1, 8.1, 6.2, 7.7, 8.9 and 14.9, 14.9, 6.3, 5.5, 9.8 for $K$=1, 2, 3, 5, 10 on Novel split1, split2 and split3. The lower the shot, the more difficult to learn a discriminative novel classifier. The significant performance gap reflects our FADI can effectively alleviate such problem even under low shot, *i.e.*, $K <= 3$. Similar improvements can be observed on the challenging COCO benchmark. As shown in Table 2, we boost TFA by 2.3, 2.4, 2.0, 1.8, 2.2, 2.4

| Association | Disentangling | Margin | nAP50 | | |
|---|---|---|---|---|---|
| | | | 1 | 3 | 5 |
| ✗ | ✗ | ✗ | 41.3 | 46.3 | 53.7 |
| ✓ | ✗ | ✗ | 42.4 | 46.8 | 55.2 |
| ✗ | ✓ | ✗ | 42.2 | 47.3 | 54.1 |
| ✓ | ✓ | ✗ | 44.9 | 50.3 | 56.8 |
| ✗ | ✗ | ✓ | 46.3 | 48.8 | 56.4 |
| ✓ | ✓ | ✓ | **50.3** | **54.2** | **59.3** |

Table 3: Effectiveness of different components of FADI.

| Margin | nAP50 |
|---|---|
| TFA | 41.3 |
| CosFace [27] | 38.9 |
| ArcFace [5] | 37.9 |
| CosFace (novel) | 44.2 |
| ArcFace (novel) | 44.3 |
| Ours | **46.3** |

Table 4: Comparison of different margin loss on the TFA baseline model.

| base / novel | bird | bus | cow | motorbike | sofa | nAP50 |
|---|---|---|---|---|---|---|
| random | person | boat | horse | aeroplane | sheep | 39.6 |
| human | aeroplane | train | sheep | bicycle | chair | 44.1 |
| visual | dog | car | horse | person | chair | 43.3 |
| top2 | dog | car | sheep | tv | diningtable | 41.2 |
| top1 | horse | train | horse | bicycle | chair | 44.3 |
| top1 w/o dup | dog | train | horse | bicycle | chair | **44.9** |

Table 5: Comparison of different assign policies. Set-specialized margin loss is not adopted in this table.

for $K$=1, 2, 3, 5, 10, 30. Besides, we also report nAP50 and nAP75, a larger gap can be obtained under IoU threshold 0.5 which suggests FADI benefits more under lower IoU thresholds.

**Comparison with State-of-the-Art Methods**   Next, we compare with other latest few-shot methods. As shown in Table 1, our method pushes the envelope of current SOTA by a large margin in shot 1, 2, 3 for novel split 1 and 3. Specifically, we outperform current SOTA by 2.5, 4.3, 2.8 and 5.6, 7.8, 1.6 for $K = 1, 2, 3$ on novel split1 and 3, respectively. As the shot grows, the performance of FADI is slightly behind FSCE [23], we conjecture by unfreezing more layers in the feature extractor, the model can learn a more compact feature space for novel classes as it exploits less base knowledge, and it can better represent the real distribution than the distribution imitated by our association. However, it is not available when the shot is low as the learned distribution will over-fit training samples.

## 4.4 Ablation Study

In this section, we conduct a thorough ablation study of each component of our method. We first analyze the performance contribution of each component, and then we show the effect of each component and why they work. Unless otherwise specified, all ablation results are reported on Novel Split 1 of Pascal VOC benchmark based on our implementation of TFA [28].

**Component Analysis**   Table 3 shows the effectiveness of each component, *i.e.*, Association, Disentangling, and Set-Specialized Margin Loss in our method. It is noted that when we study association without disentangling, we train a modified TFA model by replacing the $FC_2$ with $FC_2'$ after the association step. Since the association confuses the novel and its assigned base class, the performance of only applying association is not very significant. However, when equipped with disentangling, it can significantly boost the nAP50 by 3.6, 4.0, 3.1 for $K$=1, 3, 5, respectively. The set-specialized margin loss shows it is generally effective for both the baseline and the proposed 'association + disentangling' framework. Applying margin loss improves 'association + disentangling' by 5.4, 3,9, 2.5. With all 3 components, our method totally achieves a gain of 9.0, 7.9, 5.6.

**Semantic-Guided Association**   The assigning policy is a key component in the association step. To demonstrate the effectiveness of our semantic-guided assigning with WordNet [20], we explore different assign policies. The results are shown in Table 5. Random means we randomly assign a base class to a novel class. Human denotes manually assigning based on human knowledge. Visual denotes associating base and novel classes by visual similarity. Specifically, we regard the weights of the base classifier as prototype representations of base classes. As a result, the score prediction of novel instances on base classifier can be viewed as the visual similarity. Top1 and top2 mean the strategies that we assign each novel class to the most or second similar base classes by Eq. 1. In such

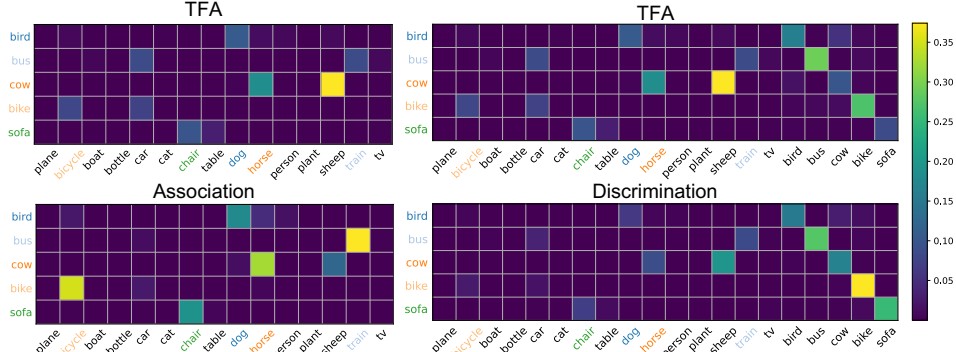

Figure 4: Score confusion matrices of different methods on Pascal VOC novel split1. The element in $i$-th row, $j$-th column represents for samples of **novel** class $i$, the score prediction on class $j$. Brighter colors indicate higher scores. If class $i$ and $j$ are the same, this indicates a more accurate score prediction. Otherwise, it indicates a heavier confusion. The font color of classes represents the association relations, *e.g.*, the associated pairs 'bird' and 'dog' have the same font color blue.

| Metric | Novel Split1 | | | Novel Split2 | | | Novel Split3 | | |
|---|---|---|---|---|---|---|---|---|---|
| | 1 | 3 | 5 | 1 | 3 | 5 | 1 | 3 | 5 |
| Visual | 43.3 | 49.3 | 56.4 | 22.5 | 37.2 | 39.3 | 31.8 | 43.1 | 50.7 |
| Semantic | **44.9** | **50.3** | **56.8** | **26.1** | **38.5** | **40.1** | **37.1** | **45.0** | **51.5** |

Table 6: Comparison of visual and semantic similarity.

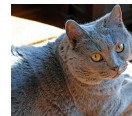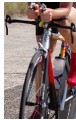

Figure 5: Examples of co-occurance

cases, one base class may be assigned to two different novel classes ("horse" is assigned to "bird" and "cow"), we remove such duplication by taking the similarity as the priority of assigning. Specifically, the base and novel classes with the highest similarity will be associated, and they will be removed from the list of classes to be associated. Then we rank the similarity of the remaining classes and choose the new association. We can learn that top1 is better than random and top2 by 4.7 and 3.1, which suggests semantic similarity has a strong implication with performance. By removing the duplication, we further obtain a 0.6 gain.

**Set-Specialized Margin Loss**  Table 4 compares our margin loss with Arcface [5] and CosFace [27]. It can be shown that directly applying these two margin losses will harm the performance. But the degeneration of performance can be reserved by only applying to samples of novel classes. This rescues Arcface from 37.9 to 44.3, Cosface from 38.9 to 44.2. Nevertheless, they are still inferior to our margin loss by 2.0. Detailed hyper-parameter study is described in the supplementary materials.

**Complementarity between Association and Discrimination**  Figure 4 shows the score confusion matrices of different methods. We can see that there exists an inherent confusion between some novel and base classes, e.g. in the left top figure, "cow" is confused most with "sheep" and then "horse". However, our association biases such confusion and enforces "cow" to be confused with its more semantic similar class "horse", which demonstrates the association step can align the feature distribution of the associated pairs. On the other hand, thanks to the discrimination step, the confusion incurred by association is effectively alleviated and overall it shows less confusion than TFA (the second column of Figure 4). Moreover, our FADI yields significantly higher score predictions than TFA, which confirms the effectiveness of disentangling and set-specialized margin loss.

**Superiority of Semantic Similarity over Visual Similarity**  Table 5 demonstrates semantic similarity works better than visual similarity. Here the weights of the base classifier as prototype representations of base classes. Thus, we take the score prediction of novel instances on base classifier as the visual similarity. However, we find it sometimes can be misleading, especially when a novel instance co-occurrent with a base instance, *e.g.*, 'cat' sits on a 'chair', 'person' rides a 'bike' as shown in Figure 5. Such co-occurrence deceives the base classifier that 'cat' is similar to 'chair' and 'bike' is similar to 'person', which makes the visual similarity not reliable under data scarcity scenarios. As shown in Table 6, when the shot grows, the performance gap between semantic and visual can be reduced by a more accurate visual similarity measurement.

# 5 Conclusion

In this paper, we propose **F**ew-shot object detection via **A**ssociation and **DI**scrimination (**FADI**). In the association step, to learn a compact intra-class structure, we selectively associate each novel class with a well-trained base class based on their semantic similarity. The novel class readily learns to align its intra-class distribution to the associated base class. In the discrimination step, to ensure the inter-class separability, we disentangle the classification branches for base and novel classes, respectively. A set-specialized margin loss is further imposed to enlarge the inter-class distance. Experiments results demonstrate that FADI is a concise yet effective solution for FSOD.

**Acknowledgements.** This research was conducted in collaboration with SenseTime. This work is supported by GRF 14203518, ITS/431/18FX, CUHK Agreement TS1712093, Theme-based Research Scheme 2020/21 (No. T41-603/20- R), NTU NAP, RIE2020 Industry Alignment Fund – Industry Collaboration Projects (IAF-ICP) Funding Initiative, and Shanghai Committee of Science and Technology, China (Grant No. 20DZ1100800).

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
