# Supplementary: Few-Shot Object Detection via Association and Discrimination

**Yuhang Cao**[1] **Jiaqi Wang**[1,2✉] **Ying Jin**[1] **Tong Wu**[1]
**Kai Chen**[2,3] **Ziwei Liu**[4] **Dahua Lin**[1,2]

[1]CUHK-SenseTime Joint Lab, The Chinese University of Hong Kong
[2]Shanghai AI Laboratory   [3]SenseTime Research
[4]S-Lab, Nanyang Technological University
{cy020,wj017,jy021,wt020,dhlin}@ie.cuhk.edu.hk
chenkai@sensetime.com   ziwei.liu@ntu.edu.sg

## 1   Appendix A: Limitation

| Method / Shot | Backbone | Novel Split 1 | | | | | Novel Split 2 | | | | | Novel Split 3 | | | | |
|---|---|---|---|---|---|---|---|---|---|---|---|---|---|---|---|---|
| | | 1 | 2 | 3 | 5 | 10 | 1 | 2 | 3 | 5 | 10 | 1 | 2 | 3 | 5 | 10 |
| LSTD [1] | VGG-16 | 8.2 | 1.0 | 12.4 | 29.1 | 38.5 | 11.4 | 3.8 | 5.0 | 15.7 | 31.0 | 12.6 | 8.5 | 15.0 | 27.3 | 36.3 |
| YOLOv2-ft [16] | YOLO V2 | 6.6 | 10.7 | 12.5 | 24.8 | 38.6 | 12.5 | 4.2 | 11.6 | 16.1 | 33.9 | 13.0 | 15.9 | 15.0 | 32.2 | 38.4 |
| †FSRW [7] | | 14.8 | 15.5 | 26.7 | 33.9 | 47.2 | 15.7 | 15.3 | 22.7 | 30.1 | 40.5 | 21.3 | 25.6 | 28.4 | 42.8 | 45.9 |
| †MetaDet [16] | | 17.1 | 19.1 | 28.9 | 35.0 | 48.8 | 18.2 | 20.6 | 25.9 | 30.6 | 41.5 | 20.1 | 22.3 | 27.9 | 41.9 | 42.9 |
| †RepMet [8] | InceptionV3 | 26.1 | 32.9 | 34.4 | 38.6 | 41.3 | 17.2 | 22.1 | 23.4 | 28.3 | 35.8 | 27.5 | 31.1 | 31.5 | 34.4 | 37.2 |
| FRCN-ft [16] | FRCN-R101 | 13.8 | 19.6 | 32.8 | 41.5 | 45.6 | 7.9 | 15.3 | 26.2 | 31.6 | 39.1 | 9.8 | 11.3 | 19.1 | 35.0 | 45.1 |
| FRCN+FPN-ft [15] | | 8.2 | 20.3 | 29.0 | 40.1 | 45.5 | 13.4 | 20.6 | 28.6 | 32.4 | 38.8 | 19.6 | 20.8 | 28.7 | 42.2 | 42.1 |
| †MetaDet [16] | | 18.9 | 20.6 | 30.2 | 36.8 | 49.6 | 21.8 | 23.1 | 27.8 | 31.7 | 43.0 | 20.6 | 23.9 | 29.4 | 43.9 | 44.1 |
| †Meta R-CNN [19] | | 19.9 | 25.5 | 35.0 | 45.7 | 51.5 | 10.4 | 19.4 | 29.6 | 34.8 | 45.4 | 14.3 | 18.2 | 27.5 | 41.2 | 48.1 |
| TFA w/ fc [15] | FRCN-R101 | 36.8 | 29.1 | 43.6 | 55.7 | 57.0 | 18.2 | 29.0 | 33.4 | 35.5 | 39.0 | 27.7 | 33.6 | 42.5 | 48.7 | 50.2 |
| TFA w/ cos [15] | | 39.8 | 36.1 | 44.7 | 55.7 | 56.0 | 23.5 | 26.9 | 34.1 | 35.1 | 39.1 | 30.8 | 34.8 | 42.8 | 49.5 | 49.8 |
| MPSR [17] | | 41.7 | - | 51.4 | 55.2 | 61.8 | 24.4 | - | 39.2 | 39.9 | 47.8 | 35.6 | - | 42.3 | 48.0 | 49.7 |
| SRR-FSD [22] | | 47.8 | 50.5 | 51.3 | 55.2 | 56.8 | **32.5** | **35.3** | 39.1 | 40.8 | 43.8 | 40.1 | 41.5 | 44.3 | 46.9 | 46.4 |
| FSCE [14] | | 44.2 | 43.8 | 51.4 | **61.9** | **63.4** | 27.3 | 29.5 | **43.5** | **44.2** | **50.2** | 37.2 | 41.9 | 47.5 | 54.6 | 58.5 |
| FADI (Ours) | | **50.3** | **54.8** | **54.2** | 59.3 | 63.2 | 30.6 | 35.0 | 40.3 | 42.8 | 48.0 | **45.7** | **49.7** | **49.1** | **55.0** | **59.6** |

Table 1: Performance (novel AP50) on PASCAL VOC dataset. This table is the same to Table 1 in the paper. We place it here for reading convenience.

| base / novel | aeroplane | bottle | cow | horse | sofa | nAP50 | | |
|---|---|---|---|---|---|---|---|---|
| | | | | | | 1 | 2 | 3 |
| semantic | **boat** | pottedplant | sheep | dog | chair | 30.6 | 35.0 | 40.3 |
| shape | **bird** | pottedplant | sheep | dog | chair | **31.5** | **35.6** | **41.6** |

Table 2: Comparison of shape similarity and semantic similarity on Pascal VOC novel split2.

As shown in Table 1, FADI achieves new state-of-the-art performance on extremely few-shot scenarios, *i.e.*, $K$=1, 2, 3 on novel split 1 and 3. However, the performance of FADI is slightly less-than-satisfactory on higher shot and novel split2. Here we analyze the possible reasons and summarize them as two limitations.

**Limitation of Feature Distribution Alignment**   As shown in Table 1, on novel split1 for $K$=5 and 10, FADI is 2.6, 0.2 lower than FSCE [14] that unfreezes more layers during finetuning when shots grow. We conjecture one possible reason is that the imitated feature distribution in the Feature

---

✉Corresponding author.

35th Conference on Neural Information Processing Systems (NeurIPS 2021).

Distribution Alignment (Section 3.2 in paper) is not the real distribution of novel classes. Specifically, by aligning the feature distribution with a well-learned base class, the Feature Distribution Alignment effectively compacts the intra-class structure of novel classes, especially when the shot is low. However, it ignores the distinctness between the novel classes and the assigned base classes, leading to a distortion of real feature representation of novel classes. With enough novel data, by unfreezing more layers in the feature extractor [14], the model learns a good feature representation for novel classes, which can better uncover the real feature distribution than our association. However, it is not available when the shot is low as the learned distribution will over-fit training samples.

**Limitation of Semantic Similarity**   As mentioned in Section 4.4 (Paragraph: Superiority of Semantic Similarity over Visual Similarity), the visual representation is not reliable under data-scarce scenarios due to the existence of co-occurrence, thus we adopt semantic as similarity measurement, which enables us to find a reasonable assigning scheme. However, it can not capture some other cues that matter to the performance, *e.g.*, shape similarity, which has been proved to be beneficial to the knowledge generalization [4]. Here we investigate the superiority of integrating the shape similarity together with semantic similarity. As shown in Table 2, by only changing one associated pair, *i.e.*, for the novel class 'aeroplane', its most semantically similar base class is 'boat', and we manually change it to the base class 'bird'. Empirically, 'bird' is more similar to 'aeroplane' from the perspective of shape. Such a simple replacement brings substantial gains which demonstrates shape similarity is better than semantic in some particular cases. We believe that a similarity measurement that incorporates both semantic and shape cues will further boost the performance of FADI.

## 2   Appendix B: Implementation Details

We implement our methods based on MMDetection [2]. Faster-RCNN [13] with Feature Pyramid Network [12] and ResNet-101 [5] are adopted as base model. To reduce the randomness of base training, we convert the base model trained by TFA [15] with Detectron2 [18] to the format of MMDetection. During the finetuning stage, we adopt the same training/testing configuration as TFA [15] for both the association stage and discrimination stage. During training, two data augmentation strategies are used, including horizontal flipping and random sizing. The network is optimized by SGD with a learning rate of $0.001$, momentum of $0.9$ and weight decay of $0.0001$. All models are trained on 8 Titan-XP GPUs with batch size 16 (2 images per GPU). The training iteration is scaled as the shot grows, specifically, the number of iterations is set to be 4000, 8000, 12000, 16000, 20000 for $K$=1, 2, 3, 5, 10, respectively. All code will be published to ensure strict reproducibility to facilitate future research.

## 3   Appendix C: Hyper-Parameter Study of Set-Specialized Margin Loss

| $\beta$ | nAP50 |
|:---:|:---:|
| 0.0 | 44.9 |
| 0.5 | 49.7 |
| 1.0 | **50.0** |
| 2.0 | 49.8 |

(a) Parameter: $\beta$

| $\gamma$ | nAP50 |
|:---:|:---:|
| 0.0 | 49.8 |
| 0.0001 | 49.8 |
| 0.001 | **50.2** |
| 0.01 | 49.4 |

(b) Parameter: $\gamma$

| $\alpha/\beta$ | nAP50 |
|:---:|:---:|
| 0.00 | 50.2 |
| 0.33 | **50.3** |
| 0.50 | 50.2 |
| 1.00 | 49.6 |

(c) Parameter: $\alpha$

Table 3: Ablation experiments for parameters $\alpha, \beta, \gamma$ on Novel Split1 Shot1.

We study the hyper-parameters, *i.e.*, $\alpha, \beta, \gamma$ adopted in set-specialized margin loss. $\alpha, \beta, \gamma$ control the magnitude of base, novel and negative margin, respectively. We found the optimality of hyper-parameters is related to the confusion level with models, hence we adopt 'association + disentangling' as the baseline framework here.

We first investigate the importance of each component in Table 3. The main performance gain comes from the novel margin (Table 3a). A higher $\beta$ can boost the classification score of novel classes, but it also has a risk of increasing false positives. Hence we introduce the base margin and negative margin to balance it. Higher $\alpha$ and $\gamma$ can yield higher scores on base and negative classes, respectively.

| $\beta$ | nAP50 |
|---|---|
| 1.00 | **50.3** |
| 0.50 | 49.9 |
| 0.33 | 49.5 |
| 0.20 | 49.0 |

(a) $K$=1

| $\beta$ | nAP50 |
|---|---|
| 1.00 | 53.7 |
| 0.50 | 54.0 |
| 0.33 | **54.2** |
| 0.20 | 53.7 |

(b) $K$=3

| $\beta$ | nAP50 |
|---|---|
| 1.00 | 58.2 |
| 0.50 | 58.8 |
| 0.33 | 58.8 |
| 0.20 | **59.3** |

(c) $K$=5

Table 4: Ablation experiments for $\beta$ on novel split1 for different shot $K$. We adopt $\alpha = \frac{1}{3}\beta, \gamma = 0.001$ for all shots here. 0.33 represents $\frac{1}{3}$.

On the contrary, it also has a suppression of novel classes. Through a coarse search, we adopt $\beta = 1.0, \alpha = \beta/3, \gamma = 0.001$ for $K = 1$, which obtains a total 5.4 gain on top of 'association + disentangling'.

Next, we explore the relation of $\beta$ and shot $K$. Intuitively, the model is easier to learn a better classifier when given more data, hence relying less on the margin loss. As shown in Table 4, when shot grows, it prefers a lower $K$. And $\beta = 1/K$ compares favorably to others. To this end, we adopt $\beta = 1/K, \alpha = \frac{1}{3}\beta$ and $\gamma = 0.001$ for different shot $K$.

# 4 Appendix D: Semantic Similarity Table on Pascal VOC

| | | | Novel Split1 | | |
|---|---|---|---|---|---|
| novel | | bird | bus | cow | motorbike | sofa |
| assigned base | | dog | train | horse | bicycle | chair |

| | | | Novel Split2 | | |
|---|---|---|---|---|---|
| novel | | aeroplane | bottle | cow | horse | sofa |
| assigned base | | boat | car | sheep | dog | chair |

| | | | Novel Split3 | | |
|---|---|---|---|---|---|
| novel | | boat | cat | motorbike | sheep | sofa |
| assigned base | | aeroplane | dog | bicycle | cow | chair |

Table 5: Complete assigning policies on Pascal VOC dataset.

# 5 Appendix E: Base Forgetting Property

Base forgetting comparison is illustrated in Table 6. As shown in this table, compared with the TFA baseline, FADI significantly improves the novel performance with just a minor performance drop on base classes. FADI outperforms MPSR (which also reports performance on both base and novel classes) on both base and novel classes with a large margin.

| Method | Base AP50 | | | Novel AP50 | | |
|---|---|---|---|---|---|---|
| | 1 | 3 | 5 | 1 | 3 | 5 |
| MPSR [17] | 59.4 | 67.8 | 68.4 | 41.7 | 51.4 | 55.2 |
| TFA [15] | **79.6** | **79.1** | **79.3** | 39.8 | 44.7 | 55.7 |
| FADI | 78.3 | 78.9 | 79.2 | **50.3** | **54.2** | **59.3** |

Table 6: Comparison of base forgetting property on Pascal VOC novel split1.

# 6 Appendix 7: Comparisons with More Co-current Methods

In the following table, we report the comparison with more co-current methods on Pascal VOC novel split1. FADI compares favorably against all of these works, which confirms the effectiveness of FADI.

| Method / Shot | Conference | Novel Split 1 | | | | |
|---|---|---|---|---|---|---|
| | | 1 | 2 | 3 | 5 | 10 |
| Zhang, Weilin, et al. [21] | CVPR2021 | 45.1 | 44.0 | 44.7 | 55.0 | 55.9 |
| Li, Yiting, et al. [11] | CVPR2021 | 40.7 | 45.1 | 46.5 | 57.4 | 62.4 |
| Fan, Zhibo, et al. [3] | CVPR2021 | 42.4 | 45.8 | 45.9 | 53.7 | 56.1 |
| Li, Aoxue, et al.* [9] | CVPR2021 | 27.7 | 36.5 | 43.3 | 50.2 | 59.6 |
| Zhang, Lu, et al. [20] | CVPR2021 | 48.6 | 51.1 | 52.0 | 53.7 | 54.3 |
| Hu, Hanzhe, et al.* [6] | CVPR2021 | 33.9 | 37.4 | 43.7 | 51.1 | 59.6 |
| Li, Bohao, et al. [10] | CVPR2021 | 41.5 | 47.5 | 50.4 | 58.2 | 60.9 |
| FADI (Ours) | NeurIPS2021 | **50.3** | **54.8** | **54.2** | **59.3** | **63.2** |

Table 7: Performance (novel AP50) on PASCAL VOC dataset. ∗ denotes average over multiple runs.

# 7 Appendix G: Broader Impact

Object detection has achieved substantial progress in the last decade. However, the strong performance heavily relies on a large amount of labeled training data. Hence a great interest is invoked to explore the few-shot detection problem (FSOD). The proposed FADI shows a great performance superiority on the current academic datasets of FSOD, but it may not be directly applicable to some realistic scenes. On the one hand, like other FSOD algorithms, the generalized performance on the novel set may be greatly affected by the concrete categories composition of the pre-training base set. Thus, the data collecting process of the base set needs to be carefully considered before using our method. On the other hand, as an emerging task, the current FSOD setting is still naive and imperfect, which limits the generalizability for the current FSOD algorithms to be applied to a more complex scenario, which may results in an unpredictable error and the degeneration of the performance.