# OpenReview forum: "Few-Shot Object Detection via Association and DIscrimination"
_NeurIPS.cc/2021/Conference — NeurIPS 2021 Poster_

### Official Review · Reviewer_zrXC · 2021-07-16

**Rating:** 6
**Confidence:** 5

**Summary:**

This paper proposes a two-step fine-tuning framework for few-shot object detection, where the first stage is an association step, associating each novel class with a base class according to their semantic similarity, and the second stage is a discrimination step, which ensure the separability between the novel classes and associated base classes. Experiments show the effectiveness of the proposed method.

**Ethical Concerns:**

No.

**Limitations And Societal Impact:**

1. If the semantic similarity between the novel class and all base class is small, there may be some problems with the association process. It is possible to choose the most similar one among the many dissimilar base classes. However, the selected base class may be not similar to the novel class. On the other hand, it is possible that the similarity between a novel class and multiple base classes is very high. In this case, there may be some problems when aligning it to one of the base classes.

2. In the association step, there may be a visual semantic gap. That is, some classes are semantically similar, but not visually similar. This problem is also common in zero-shot learning and zero-shot object detection [1]. However, the authors didn’t consider this issue.

[1] C. Yan, Q. Zheng, X. Chang, M. Luo, C. -H. Yeh and A. G. Hauptman, "Semantics-Preserving Graph Propagation for Zero-Shot Object Detection," in IEEE Transactions on Image Processing, vol. 29, pp. 8163-8176, 2020, doi: 10.1109/TIP.2020.3011807.

3. The experimental setting of this paper is different from the previous FSOD works. This work introduces the semantic information of the category, which is equivalent to having additional information input. There are some works in few-shot learning [2], and the introduction of semantic information can improve performance by a large margin. However, the performance improvement brought by semantic information in this paper is limited (in Tab.3, less than 1 point in 1/3-shot)，this makes the effectiveness of the method questionable.

[2] Chen Xing, Negar Rostamzadeh, Boris N Oreshkin and Pedro O Pinheiro, "Adaptive cross-modal few-shot learning", Advances in Neural Information Processing Systems, pp. 4848-4858, 2019.


**Main Review:**

The paper proposes a new framework for few-shot object detection and achieves good results on some indicators. Generally, the paper is well organized and written.

However, there are obvious limitations with this paper:
1. Feature distribution alignment module is interesting and novel. However, the proposed set-specialized margin loss has limited innovation. The loss is very similar to cos similarity, which is also used in TFA. Each line of W in Eq.5 can be regraded as a prototype of a category. The main innovation of the proposed loss function lies in that the weights of the novel classes are increased by hyperparameter $\beta$, and the weight of the background class is reduced by hyperparameter $\gamma$. It's more like a technical trick. In line 198-201, the authors explained the motivation of reweighing. As can be seen from Appendix C, when $\alpha/\beta $ increases, the AP50 improves little (only from 49.6 to 50.3). Similarly, for the adjustment of gamma, the effect on the result is relatively small. Therefore, the validity of this technique is questionable. Moreover, the authors did not do experiment for $\gamma$ = 1 in Table 4 (b) of Appendix C.

2. In experiments, comparisons with the latest SOTAs are missed. Following are
some papers in CVPR2020 have not been compared:
[1] Fan, Qi, et al. "Few-shot object detection with attention-RPN and multi-relation detector." Proceedings of the IEEE/CVF Conference on Computer Vision and Pattern Recognition. 2020.
[2] Yang, Yukuan, et al. "Restoring Negative Information in Few-Shot Object Detection." Advances in Neural Information Processing Systems 33 (2020): 3521-3532.
[3] Yang, Ze, et al. "Context-transformer: tackling object confusion for few-shot detection." Proceedings of the AAAI Conference on Artificial Intelligence. Vol. 34. No. 07. 2020.

In addition, comparing the papers in CVPR 2021, the performance of the proposed method is not promising:
[4] Zhang, Weilin, and Yu-Xiong Wang. "Hallucination Improves Few-Shot Object Detection." Proceedings of the IEEE/CVF Conference on Computer Vision and Pattern Recognition. 2021.
[5] Li, Yiting, et al. "Few-Shot Object Detection via Classification Refinement and Distractor Retreatment." Proceedings of the IEEE/CVF Conference on Computer Vision and Pattern Recognition. 2021.
[6] Fan, Zhibo, et al. "Generalized Few-Shot Object Detection without Forgetting." Proceedings of the IEEE/CVF Conference on Computer Vision and Pattern Recognition. 2021.
[7] Li, Aoxue, and Zhenguo Li. "Transformation Invariant Few-Shot Object Detection." Proceedings of the IEEE/CVF Conference on Computer Vision and Pattern Recognition. 2021.
[8] Zhang, Lu, et al. "Accurate Few-Shot Object Detection With Support-Query Mutual Guidance and Hybrid Loss." Proceedings of the IEEE/CVF Conference on Computer Vision and Pattern Recognition. 2021.
[9] Hu, Hanzhe, et al. "Dense Relation Distillation with Context-aware Aggregation for Few-Shot Object Detection." Proceedings of the IEEE/CVF Conference on Computer Vision and Pattern Recognition. 2021.
[10] Li, Bohao, et al. "Beyond Max-Margin: Class Margin Equilibrium for Few-shot Object Detection." Proceedings of the IEEE/CVF Conference on Computer Vision and Pattern Recognition. 2021.

3. Line 181:  Where is $\widetilde{W}^B_{asso}$ from? As I understand, in the association step, we can obtain $\widetilde{W}^N_{asso}$, which is the intermediate structure that follows $\phi(;\widetilde{W}^B_{pre})$ (feature extractor). $\widetilde{W}^N_{asso}$ is used to perform feature distribution alignment for novel classes, but for base classes, it doesn’t need to perform feature distribution alignment. So, what does $\widetilde{W}^B_{asso}$ mean? In line 181-182, it’s said that it comes from the association step, but I cannot find the variable in the association step.


**Time Spent Reviewing:**

4

---

> ### Author Response · Authors · 2021-08-10
> **Response to reviewer**
>
> **Q1: Feature distribution alignment module is interesting and novel...**
>
> As in lines 192-193 and Eq. 5, we follow the convention (*e.g.*, TFA) to adopt cosine similarity as classification logits. In TFA, the logits are then adopted to calculate the cross-entropy loss for classification.  However, in FADI, we further adopt a Set-Specialized Margin Loss (Eq. 6 and Eq. 7) which helps to maximize the margin of decision boundary between different classes.
> Thus, our contributions are in Eq. 6 and Eq. 7 rather than in Eq. 5 (cosine similarity).
> The other key contribution is to provide set-specific treatments towards different sets of classes.
> As in Table 4 (row 2 and row 3) in the main text, traditional margin losses (Cosine Face, ArcFace) are not directly applicable because they treat different sets of samples equally.
> And there exists an inherent bias that the classifier tends to predict higher scores on base classes, which makes the optimization of margin loss on novel classes difficult.
> Furthermore, the number of background samples dominates the training samples, which makes the optimization on novel classes even more challenging.
> To ease the difficulty of optimization, we divide the samples into different sets and propose to handle it in a re-weighting manner which is simple yet effective.
>
> Here we report more experiments to better reflect the effect of re-weighting on base and negative margin. As in the table below, equally treating different sets of samples will harm the performance, *i.e.*, setting $\alpha, \beta, \gamma$ to be all 1 on shot 1, all 1 or 1/3 on shot 3, and all 1 or 1/5 on shot 5. To ease the difficulty of the optimization on novel classes, removing base and negative margin ($\alpha=0, \gamma=0$) can greatly boost performance. Moreover, applying appropriate margin on base and negative classes ($\beta$ = $1/K, \alpha=1/3\beta, \gamma=0.001$) can further boost the performance, especially in higher shots (boost 2.2 for $K$=3 and 2.1 for $K$=5).
>
> Moreover, $\gamma$ is explored in Table 3 (b) of Appendix C, and sorry for missing the setting with $\gamma$ = 1 in this table. The result with $\gamma$ = 1.0 is 45.7, which is inferior to $\gamma$ = 0.001.
>
> | Shot K | $\alpha$ | $\beta$ | $\gamma$ | nAP50 |
> |-|:-:|:-:|:-:|:-:|
> | K=1    | 1        | 1       | 1        | 43.8  |
> |        | 0        | 1       | 0        | 49.8  |
> |        | 1/3.     | 1       | 0.001    | **50.3**  |
> | K=3    | 1        | 1       | 1        | 44.9  |
> |        | 1/3.     | 1/3.    | 1/3.     | 50.6  |
> |        | 0        | 1/3.    | 0        | 52.6  |
> |        | 1/9.     | 1/3.    | 0.001    | **54.8**  |
> | K=5    | 1        | 1       | 1        | 48.8  |
> |        | 1/5.     | 1/5.    | 1/5.      | 55.9  |
> |        | 0        | 1/5.    | 0        | 57.2  |
> |        | 1/15.   | 1/5.    | 0.001    | **59.3**  |
>
>
> **Q2: In experiments, comparisons with the latest SOTAs are missed...**
>
> Thanks for your reminder. In the submission, we did not compare these papers for the following reasons. For [1], it does not report the performance on Pascal VOC and uses different train/val set compared to the convention on COCO, *i.e.*, the protocol that proposed in Kang, Bingyi, et al. "Few-shot object detection via feature re-weighting".  For [2], we notice it uses a stronger backbone (ResNet-101 with DCN) and some other tricks like OHEM, SyncBN. Thus, it is unfair to compare with FADI. For the papers in CVPR2021, they are officially published after the submission deadline of NeurIPS 2021, hence we missed them in the paper.
>
> In the following table, we report the comparison with [2], [3], [4], [5], [6], [7], [8], [9], [10] on Pascal VOC split1. FADI compares favorably against all of these works, which confirms that FADI achieves SOTA on shot $K$=1, 2, 3 for split 1. For [1], with the same setting on COCO shot10, nAP of FADI is 12.1 and outperforms [1] (11.1) by 1.0. We will add comparisons with these papers in the final version.
>
> |      Method      | Conference |   1  |   2  |   3  |   5  |  10  |
> |:-:|:-:|:-:|:-:|:-:|:-:|:-:|
> |  Yang, Yukuan[2] |  NIPS2020  | 37.8 | 40.3 | 41.7 | 47.3 | 49.4 |
> |    Yang, Ze[3]   |  AAAI2020  | 34.2 |   -  |   -  | 44.2 |   -  |
> | Zhang, Weilin[4] |  CVPR2021  | 45.1 | 44.0 | 44.7 | 55.0 | 55.9 |
> |   Li, Yiting[5]  |  CVPR2021  | 40.7 | 45.1 | 46.5 | 57.4 | 62.4 |
> |   Fan, Zhibo[6]  |  CVPR2021  | 42.4 | 45.8 | 45.9 | 53.7 | 56.1 |
> |   Li, Aoxue[7]   |  CVPR2021  | 27.7 | 36.5 | 43.3 | 50.2 | 59.6 |
> |   Zhang, Lu[8]   |  CVPR2021  | 48.6 | 51.1 | 52.0 | 53.7 | 54.3 |
> |   Hu, Hanzhe[9]  |  CVPR2021  | 33.9 | 37.4 | 43.7 | 51.1 | 59.6 |
> |   Li, Bohao[10]  |  CVPR2021  | 41.5 | 47.5 | 50.4 | 58.2 | 60.9 |
> |   FADI (Visual)  |      -     | 48.6 | 53.7 | 53.0 | 58.7 | 62.2 |
> |  FADI (Semantic) |      -     | **50.3** | **54.8** | **54.2** | **59.3** | **63.2** |
>
>
> **Q3: Line 181: Where is $W_{asso}^B$ from...**
>
> $W_{asso}^B$ is the weight of $FC_2$ loaded from the pre-trained model on base classes. $FC_2$ and $FC_2'$ have the same structure as mentioned in lines 180-182. $FC_2$ loads the weights $W_{asso}^B$ from the pre-trained model on base classes, which is responsible for the prediction of base classes. $FC_2'$ loads the weights $W_{asso}^N$ from the model of association, which is responsible for the prediction of novel classes. We notice the name of $W_{asso}^B$ may be misleading, thus we may rename it as $W_{pretrain}^B$ for clarification.
>
>
> **Q4: If the semantic similarity between the novel class and all base class is small...**
>
> Thanks for pointing the problems.
>
> For the former, we agree that this undesirable situation may happen in some cases. However, if the base classes are diverse, most novel classes are likely to have some similarities with base classes. And the association step will benefit these novel classes. The extensive experiments on both Pascal VOC and COCO show the strong performance of FADI, it reveals that the superiority of the association step even without explicitly avoiding the potential undesirable association between base and novel classes. Furthermore, we also explore to manually remove the dissimilar association and evaluate the performance. As shown in the following table, we explore to exclude the dissimilar association between bottle and car.
> Results show that it brings negligible benefits on the performance.
>
> For the latter, actually it is not a limitation but a problem that we are solving. As in Figure 1 in the paper, if a novel class is similar to multiple base classes, it will induce a scattered intra-class structure (Fig 1 (a), cow is similar to sheep and horse). Our association compacts the intra-class structure by associating that class to a single well-learned base classes (Fig 1 (b), cow is associated with sheep). Figure 3 validates the effectiveness of association. In Fig. 3 (a) cow (yellow) is mixed with horse (purple) and sheep (pink). In Fig. 3 (b), association separates cow from sheep and associates it with horse.
>
> | aeroplane | bottle |  cow  | horse |  sofa |  K=1 |  K=3 |  K=5 |
> |:-:|:-:|:-:|:-:|:-:|:-:|:-:|:-:|
> |    boat   |   car  | sheep |  dog  | chair | 30.6 | **40.3** | **42.8** |
> |    boat   |  **none**  | sheep |  dog  | chair | **30.9** | 39.8 | 42.4 |
>
>
> **Q5: In the association step, there may be a visual semantic gap...**
>
> Thanks for pointing out this problem. And we will cite this mentioned paper.
> As discussed in Sec 4.4 in the main text: Superiority of Semantic Similarity over Visual Similarity, directly computing visual similarity is not accurate due to co-occurrence, especially when shot is low. Hence we use semantic similarity as a workaround to the limitation of directly computing the visual similarity. Extensive experiments on Pascal VOC and COCO confirm the robustness of semantic similarity.
> Nevertheless, a better similarity measurement can promote the performance of association. As in Table 2 of Appendix A, the aeroplane is associated with the boat due to their semantic similarity (both are vehicles). However, it is better to associate the aeroplane with the bird due to their similar shapes. One association pair replacement, *i.e.*, (boat, aeroplane) $\rightarrow$ (bird, aeroplane), increases the performance by 0.6$\sim$0.9.
> Thus, a similarity measurement that incorporates both semantic and visual cues may further boost the performance of FADI.
> How to find a more accurate approximation to the visual representation has beyond the scope of this paper. We leave it as a future research direction.
>
> **Q6: The experimental setting of this paper is different from the previous FSOD works...**
>
> Semantic information used in FADI is embedded in WordNet and Lin Similarity (Eq. 1), which serves as a prior to the association step. It introduces zero cost to the whole framework. Moreover, FADI does not utilize semantic information during testing.
>
> As in lines 255-257, the association will confuse the novel and its assigned base classes. As in the following table (Reviewers may also refer to Table 3 in the main text), the performance of only applying association (row 2) is not very significant. However, when equipped with disentangling (row 4), it can significantly boost the nAP50 by 2.7, 3.0, 2.7 for K=1, 3, 5, respectively.
>
> Moreover, the proposed association step is not limited to semantic similarity. For example, we may replace it with visual similarity as in Line 289-291. As shown in the table in \bd{Q2}, although FADI with visual similarity is inferior to FADI with semantic similarity, it still achieves competitive performance compared to previous SOTA methods.
>
> And we will cite this mentioned paper.
>
> | Association  | Disentangling | 1    | 3    | 5    |
> |:-:|:-:|:-:|:-:|:-:|
> |              |               | 41.3 | 46.3 | 53.7 |
> | $\checkmark$ |               | 42.4 | 46.8 | 55.2 |
> |              | $\checkmark$  | 42.2 | 47.3 | 54.1 |
> | $\checkmark$ | $\checkmark$  | **44.9** | **50.3** | **56.8** |

---

> > ### Comment · Reviewer_zrXC · 2021-09-01
> > **Score update**
> >
> > Thanks for the  explanation. The authors have adressed most of the reviewer's concerns. Therefore, the reviewer decide to update the score from 5 to 6.

---

### Official Review · Reviewer_oVR6 · 2021-07-17

**Rating:** 8
**Confidence:** 4

**Summary:**

The paper proposes a novel approach for few-shot object detection by first assigning a novel (few-shot) class to a base category (Association) and then disentangling the classification of base and novel classes (Discrimination). After the first step of base training, the proposed approach assigns each novel class to a base class using the WordNet hierarchy. Then, the method aligns the feature distribution of the novel class to the associated base class. Finally, the method proposes to disentangle the classification branches of base and novel classes to remove confusion between base and novel classes.

**Limitations And Societal Impact:**

The authors have adequately addressed the limitations and potential negative social impact of their work but only in the supplementary material and not in the main text of the paper.

**Main Review:**

The proposed approach is highly intuitive and has been well described by the authors. I believe that this idea is novel and is a strong contribution to the field. The authors have also demonstrated the effectiveness of the proposed approach through strong results on the PASCAL VOC and MSCOCO datasets in the few-shot setting.

I have a few questions/suggestion about the paper which the authors should address in their response.

1. In Table 1, the authors show good improvements for low-shot learning in Split 1 and Split 3. Why is the performance not so good for 5 and 10 shots for Split 1? 5-shot seems to be particularly low compared to FSCE. Why is this the case? Was the performance of FSCE somehow artificially too high in this case or is the performance of FADI artificially too low? Further, why is there no similar improvement for Split 2? How is Split 2 different from Splits 1 and 2?

2. For Table 5, do the same observations hold for larger datasets like MSCOCO? Do these observation change with change in dataset?

**Time Spent Reviewing:**

4

---

> ### Author Response · Authors · 2021-08-10
> **Response to reviewer**
>
> **Q1: The proposed approach is highly intuitive and has been well described by the authors...**
>
> Thanks for your appreciation.
>
>
> **Q2: In Table 1, the authors show good improvements for low-shot learning in Split 1 and Split 3...**
>
> We also observed this phenomenon. And you may refer to Appendix A: limitations for some related discussion. Here we summarize the discussion as follows.
>
> 1. **Unsatisfactory performance on shot 5, 10**. FADI aligns the feature distribution of the novel class with its assigned base class. It effectively compacts the intra-class structure of novel classes, especially when the shot is low. But we should notice that the imitated feature distribution is not the real distribution of novel classes, leading to a gap towards the real feature representation of novel classes. Instead, FSCE unfreezes more layers and directly learns the feature distribution from novel data. The learned distribution may be better than the distribution imitated by our association step when the shot is high. But the learned distribution performs worse than our method under data-scarce cases (low shot).
>
> 2. **Unsatisfactory performance on split 2**. On the one hand, in different splits, the base and novel classes are different. We observe that the similarity between base and novel classes in split2 is lower than split1 and split3, which makes it difficult to align the feature distribution during the association step in FADI. On the other hand, the adopted semantic similarity is not an optimal similarity measurement for split2. As shown in Table 2 in Appendix A, airplane and bird share high shape similarity. One association pair replacement, *i.e.*, (boat, airplane) $\rightarrow$ (bird, airplane), increases the performance by 0.6$\sim$0.9. However, shape similarity is hard to be captured by semantic similarity measurement,
> which may limit the performance of FADI in some cases.
>
> 3. **5-shot seems to be particularly low compared to FSCE**.
> As previously discussed, the design of FSCE unfreezes more components and has some natural benefits with high shot settings (e.g., 5 shot, 10 shot). In split 1 and split 2, FSCE outperforms FADI on high shots (e.g., 5 shot, 10 shot), while FADI outperforms FSCE in split 3 on all shots. In split 1, we observe that the performance of FSCE unstably increase by 10.5 from shot 3 to shot 5, while only 1.5 from shot 5 to shot 10. However,
> FADI has relatively stable performance gains by 5.1 and 3.9 from shot 3 to shot 5 and from shot 5 to shot 10, respectively. As a result, the FSCE outperforms FADI by 2.6 on shot 5 while only 0.2 on shot 10.
> We think that these unstable performance gap changes naturally comes with few shot setting, because there are only few training samples of novel classes. Thus the changes of training samples will heavily affect the performance.
> And since FSCE unfreezes more components compared to FADI, this unstable performance change seems to become more likely to happen.
>
>
> **Q3: For Table 5, do the same observations hold for larger datasets like MSCOCO?...**
>
> Below, we present the comparison of different assigning schemes on COCO dataset for shot $K$=1, set-specialized margin loss is not adopted here. Similar to Table 5, random means randomly assign a base class to a novel class. Human means manually assign by human knowledge. As shown in the table below, randomly assign has no gain compared with the TFA baseline, assigning by semantic similarity outperforms visual similarity by 0.6, but 0.2 lower than the human assigning scheme. Though COCO dataset has more classes than VOC, the performance gain aligns with our early conclusion: an appropriate similarity measurement is a key factor and semantic is better than visual.
>
> | Polices | TFA | Random | Human | Visual | Semantic |
> |:-------:|:---:|:------:|:-----:|:------:|:--------:|
> |   nAP   | 3.4 |   3.3  |  **4.8**  |   4.0  |    4.6   |
>
>
> **Q4: The authors have adequately addressed the limitations and potential negative social impact of their work but only in the supplementary material and not in the main text of the paper.**
>
> Thanks for your suggestions. We will re-organize the content and move the discussion of limitations to the main text.

---

### Official Review · Reviewer_VAwp · 2021-07-18

**Rating:** 6
**Confidence:** 4

**Summary:**


The paper proposes a method for few shot object detection. The key idea is to leverage the similarity of a novel object class with an existing base class, and then train a local classifier for better discrimination. The class similarity is based on semantic word similarity between the classes’ textual names and the new class to base class mapping is an injective mapping. Once the assignment is calculated the feature network is trained so that the features of the new class align with its corresponding base class. Finally further discriminator layers are used which are then trained to classify base and new classes distinctly. Empirical results are shown on PASCAL VOC 2007 2012 and COCO datasets, where the proposed method outperforms existing methods especially in very low shot settings.


**Main Review:**

The main motivation seems to be that if you fine-tune with the less amount of data, then the local optima of the resulting optimization might not be good. So the idea is to constrain the optimization region with priors. The first being that the feature network should be highly biased to keep similar classes in the same locality in both base and new classes. Second one being that the classifiers for the similar base and novel classes should be in some sense trained locally.

The colors used in fig4 could be better, I had to look carefully to see that book and cow are two different colors and not different weights of different colors (first reflex is to associate light color with small weight, on which I had to remind myself that the association is binary, right?). Use contrasting colors. And perhaps for the non matched classes have highlights or underlines or some other decoration to make it easy to perceive.

Also in fig4 score confusion matrix is shown, but what exactly is score confusion is not defined? It is also not very clear why the left column on shows base classes in the x axis which the right col has both base and new ones.

In table 5 association comparison with visual features derived similarity should also be compared with the semantic text similarity. It would be interesting to see how much worse is that.

The tab6 argument seems weak as in one shot situation, we would expect the variance to be very high, and the numbers to flicker a lot depending on the splitting. With even a slight increase in number of examples the visual and textual similarities seem to be similar.

The AP is reported only on novel classes (nAP). As experienced with zero and few shot classification works, there is usually a tradeoff between novel class only performance and whole set performance. I understand that practitioners may be only interested in new class performances, but academically we should be aware of how the performances are suffering for old classes, and all classes taken together. Since those metrics should be easy to compute with the new model, I would strongly encourage them to be reported (even if only for the current method, as the older methods might not have reported them. I see that TFA has reported that on base and novel separately). Since the method here is advocates strong local learning, it might be alleviating confusion for new classes, but might be increasing the same for old classes. A discussion would also be appreciated.


**Time Spent Reviewing:**

3

---

> ### Author Response · Authors · 2021-08-10
> **Response to reviewer**
>
> **Q1: The main motivation seems to be that if you fine-tune with the less amount of data, then the local optima of the resulting optimization might not be good...**
>
> It is a good explanation of the motivation of FADI from the other perspective. Thanks for your comments.
>
>
> **Q2: The colors used in fig4 could be better, I had to look carefully to see that book and cow are two different colors and not different weights of different colors...**
>
> Thanks for your suggestions. We will rework Fig. 4 to make it more clear. Besides the caption of Fig. 4, here we provide more explanations of this figure.
> Font color of the class name indicates association relation, *e.g.*, bird and dog are both blue, which means that novel class bird is associated with base class dog by semantic similarity.
> For an element on $i$-th row, $j$-th column on each sub-figure, it denotes that the samples belong to novel class $i$, and we show their averaged prediction scores on class $j$. For example, in the top left sub-figure, the color of the cell (cow, sheep) is very bright, which means the classifier tends to predict cow as sheep. It shows that there exists a heavy confusion between cow and sheep. In the bottom right sub-figure, the color of the cell (bike, bike) is very bright, which means the classifier predicts the correct label for bike.
>
> We only draw base classes on the x-axis in the two sub-figures on the left because we want to show association can help to align the feature distribution (*i.e.*, increase the classification confusion) between the novel classes with their associated base classes. For the two sub-figures on the right, we want to show 1) discrimination can reduce the confusion between base and novel classes, 2) discrimination can predict higher scores on correct labels. Thus, we draw all classes on the x-axis.
>
>
>
> **Q3: In table 5 association comparison with visual features derived similarity should also be compared with the semantic text similarity...**
>
> The comparison between semantic and visual is important, thus we put it in Sec. 4.4, Paragraph: Superiority of Semantic Similarity over Visual Similarity. Here we add a more concrete assigning scheme comparison. Semantic (44.9) outperforms visual (43.3) by 1.6, and the main difference is that semantic assigns bicycle to motorbike but visual assigns person to motorbike. As stated in the paper (Line 291 - 294), the visual representation can be misleading, especially when a novel instance co-occurrent with a base instance (person rides a bicycle). Although the visual representation can be calibrated with more samples (In Table 6 of the main text, the gap between semantic and visual decreases as the shot grows), there is still a performance gap between visual and semantic similarity on a higher shot (50.7 v.s. 51.5 on shot 5 as in Table 6).
>
> | base/novel | bird | bus   | cow   | motorbike | sofa  | nAP50 |
> |------------|:----:|:-----:|:-----:|:---------:|:-----:|:-----:|
> | visual     | dog  | car   | horse | person    | chair | 43.3  |
> | semantic   | dog  | train | horse | bicycle   | chair | **44.9**  |
>
>
> **Q4: The tab6 argument seems weak as in one shot situation, we would expect the variance to be very high, and the numbers to flicker a lot depending on the splitting...**
>
> To prevent the experimental variance, besides experiments on split 3, here we also report the comparison between visual similarity and semantic similarity on split1 and split2.
> As shown in the table, semantic similarity is consistently better than visual similarity by 1.6, 1.0, 0.4 and 3.6, 1.3, 0.8 for $K$=1, 3, 5 on split1 and split2, respectively.
>
> | split 1  | 1    | 3    | 5    | split 2 | 1    | 3    | 5    |
> |----------|:----:|:----:|:----:| ----------|:----:|:----:|:----:|
> | visual   | 43.3 | 49.3 | 56.4 | visual    | 22.5 | 37.2 | 39.3 |
> | semantic | **44.9** | **50.3** | **56.8** | semantic| **26.1** | **38.5** | **40.1** |
>
>
> **Q5: The AP is reported only on novel classes (nAP)...**
>
> Thanks for your suggestion. Base forgetting comparison is illustrated in the following table. As shown in this table, compared with the TFA baseline, FADI significantly improves the novel performance with just a minor performance drop on base classes.  FADI outperforms MPSR (which is reported on both base and novel classes in their paper) on both base and novel classes with a large margin.
>
> | Base AP50 |   1  |   3  |   5  | Novel AP50 |   1  |   3  |   5  |
> |-----------|:----:|:----:|:----:|------------|:----:|:----:|:----:|
> | MPSR      | 59.4 | 67.8 | 68.4 | MPSR       | 41.7 | 51.4 | 55.2 |
> | TFA       | **79.6** | **79.1** | **79.3** | TFA        | 39.8 | 44.7 | 55.7 |
> | FADI      | 78.3 | 78.9 | 79.2 | FADI       | **50.3** | **54.2** | **59.3** |

---

### Official Review · Reviewer_DsGJ · 2021-07-19

**Rating:** 6
**Confidence:** 5

**Summary:**

Durning ﬁne-tuning of few-shot object detection, a novel class may implicitly leverage the 8 knowledge of multiple base classes to construct its feature space, which induces a scattered feature space, hence violating the inter-class separability. To overcome these obstacles, we propose a two-step ﬁne-tuning framework, Few-shot object detection via Association and DIscrimination (FADI), which builds up a discriminative feature space for each novel class with two integral steps.

**Main Review:**

I am in general positive towards this paper, as the method has convincing motivation, and is presented very clearly.

I still have a few comments:

Authors claim that existing SOTA few-shot detection works all employ a finetuning strategy, this is not appropriate. There are still several works focusing on the meta-learning mode, for instance, RepMet [13] and NP-RepMet [NIPS2020].

For the class similarity measure, authors choose WordNet over over visual similarity.  From my viewpoint, visual similarity also has advantages that WordNet does not. For instance, an airplane and a bird can only be similar visually.

For the feature distribution alignment, what's the motivation to implement it this way ? Have you tried some other ways, for instance, gradient reversal layer ?

The method performs SOTA on low-shots in most scenarios, yet not in the second split of VOC. Any discussions and observations on this ?
Also, have authors tried to integrate their methods into FCSE to see if they could further achieve SOTA on rather higher shots ?


**Time Spent Reviewing:**

3 hours

---

> ### Author Response · Authors · 2021-08-10
> **Response to reviewer**
>
> **Q1: Authors claim that existing SOTA few-shot detection works all employ a finetuning strategy, this is not appropriate...**
>
> Thanks for your reminder. The meta-learning paradigm also achieved good results recently. We will revise the statement and cite the mentioned papers.
>
>
> **Q2: For the class similarity measure, authors choose WordNet over over visual similarity...**
>
> It is a great question. And we have some corresponding observations and discussions in Appendix A: Limitation of Semantic Similarity. Briefly speaking, experimental results show that semantic similarity usually works better than visual Similarity due to co-occurrence (please refer to Sec 4.4 in the main text: Superiority of Semantic Similarity over Visual Similarity). However, for some specific pairs of classes, visual similarity will work better than semantic similarity.
> As shown in Table 2 in Appendix A, the airplane (aeroplane) is associated with the boat due to their semantic similarity (both are vehicles). However, the airplane (aeroplane) will be associated with the bird by visual similarity due to their similar shapes. One association pair replacement, *i.e.*, (boat, airplane) $\rightarrow$ (bird, airplane), increases the performance by 0.6$\sim$0.9. Thus, a similarity measurement that incorporates both semantic and visual cues may further boost the performance of FADI. We leave the integration of visual and semantic measurement as a future research direction.
>
>
>
> **Q3: For the feature distribution alignment, what's the motivation to implement it this way...**
>
> Given a base class $A$ with label $y_A$, a novel class $B$ with label $y_B$, and a well trained classifier $f(\cdot; \widetilde{W}_A)$ for class $A$, our goal is to align the feature distribution of $B$ to $A$. By freezing the classifier $f(\cdot; \widetilde{W}_A)$ and changing the label of $B$ to $y_A$, in order to minimize the classification loss, the gradients will force the feature distribution of $B$ to shift towards $A$.
> If the classifier $f(\cdot; \widetilde{W}_A)$ classifies $B$ as $y_A$, it means the feature distribution of $A$ and $B$ are aligned and achieves our goal.
>
> Here we report the performance that replaces the feature distribution alignment in FADI by gradient reversal layer (GRL) on VOC split1. Specifically, we attach an extra classifier that needs to distinguish a sample between base and novel class. During training, we unfreeze $FC_2$ and the classifiers. The GRL classifier is only trained on positive samples of associated classes. The reversal of gradients happens after $FC_2$ and before the GRL classifier. As a result, the reversed gradients will mitigate the difference of the feature representation between base and novel classes after $FC_2$. As shown in the table below,
> the GRL helps to improve the baseline performance, but our FADI is still much better across different shot settings. Here the baseline means the TFA with our proposed Set-Specialized Margin Loss, since we also adopt
> this margin loss in GRL for a fair comparison with FADI.
>
> | Shot | 1    | 3    | 5    | 10   |
> |------|------|------|------|------|
> | Baseline | 46.3 | 48.8 | 56.4 | 58.5 |
> | GRL  | 47.1 | 52.5 | 57.1 | 60.5 |
> | FADI | **50.3** | **54.2** | **59.3** | **63.2** |
>
>
> **Q4: The method performs SOTA on low-shots in most scenarios, yet not in the second split of VOC...**
>
> We observed this phenomenon. And some related discussions can be found in Appendix A: limitations.
> In summary, on the one hand, in different splits, the base and novel classes are different. We observe that the similarity between base and novel classes in split2 is lower than split1 and split3, which makes it difficult to align the feature distribution during the association step in FADI. On the other hand, the adopted semantic similarity is not an optimal similarity measurement for split2. As shown in Table 2 in Appendix A, airplane and bird share high shape similarity. One association pair replacement, *i.e.*, (boat, airplane) $\rightarrow$ (bird, airplane), increases the performance by 0.6$\sim$0.9. However, shape similarity is hard to be captured by semantic similarity measurement, which may limit the performance of FADI in some cases.
>
> Here we report the performance by integrating FADI with FSCE on split2. We remain the association step the same but only combine FADI with FSCE during the discrimination step. We follow the same hyper-parameters with FSCE. As shown in the following table, FSCE + FADI outperforms FSCE across all shots on split 2.
>
> | Shot      | 1    | 2    | 3    | 5    | 10   |
> |-----------|------|------|------|------|------|
> | FSCE      | 27.3 | 29.5 | 43.5 | 44.2 | 50.2 |
> | FADI      | 30.6 | 35.0 | 40.3 | 42.8 | 48.0 |
> | FADI+FSCE | **31.6** | **38.2** | **43.8** | **44.8** | **50.6** |

---

### Decision · Program_Chairs · 2021-09-27

**Decision:**

Accept (Poster)

**Comment:**

The paper proposes an alternative to fine-tuning for few-shot object detection, by instead associating each new class to an existing base class and then training with a max margin loss between this new class from the base class. The idea is simple, its implementation probably still leaves some performance on the table (e.g. visual similarity vs. WordNet similarity). The experimental validation is strong, in particular in term of AP on new classes on standard benchmarks.

The reviewers agree that the paper is mostly clear and convincing. The authors addressed some of the limitations of the paper in the author response, in particular in term of comparisons with the literature. I recommend acceptance as a poster.